# High Potency of SN-38-Loaded Bovine Serum Albumin Nanoparticles Against Triple-Negative Breast Cancer

**DOI:** 10.3390/pharmaceutics11110569

**Published:** 2019-11-01

**Authors:** Hsin-Che Lin, Chih-Hung Chuang, Meng-Hsuan Cheng, Yu-Chih Lin, Yi-Ping Fang

**Affiliations:** 1School of Pharmacy, College of Pharmacy, Kaohsiung Medical University, Kaohsiung 80708, Taiwan; rowrowrowingboat@hotmail.com; 2Department of Medical Laboratory Science and Biotechnology, College of Health Sciences, Kaohsiung Medical University, Kaohsiung 80708, Taiwan; a4132600@kmu.edu.tw; 3Drug Development and Value Creation Research Center, Kaohsiung Medical University, Kaohsiung 80708, Taiwan; 4Division of Pulmonary and Critical Care Medicine, Department of Internal Medicine, Kaohsiung Medical University Hospital, Kaohsiung 80708, Taiwan; cmhkmu@gmail.com; 5School of Medicine, College of Medicine, Kaohsiung Medical University, Kaohsiung 80708, Taiwan; 6Department of Respiratory Therapy, College of Medicine, Kaohsiung Medical University, Kaohsiung 80708, Taiwan; 7Department of Environmental Engineering and Health, Yuanpei University of Medical Technology, Hsinchu 80708, Taiwan; yuchihlin@mail.ypu.edu.tw; 8Department of Medical Research, Kaohsiung Medical University Hospital, Kaohsiung 80708, Taiwan; 9Regenerative Medical and Cell Therapy Center, Kaohsiung Medical University, Kaohsiung 80708, Taiwan

**Keywords:** triple-negative breast cancer (TNBC), albumin nanoparticles, irinotecan, 7-ethyl-10-hydroxycamptothecin (SN-38)

## Abstract

Triple-negative breast cancer (TNBC) is an aggressive type of breast cancer with a worse prognosis than other types. There are currently no specific approved treatments for TNBC. Albumin is a promising biomimetic material that may be fabricated into nanoparticles to possibly exert passive effects on targeted tumors. Irinotecan has been extensively used in clinical settings, although a high dosage is required due to its low efficiency of conversion into the active metabolite SN-38, also known as 7-ethyl-10-hydroxy-camptothecin. The aim of this work was to optimize SN-38-loaded bovine serum albumin nanoparticles (sBSANPs) and evaluate their potency against TNBC. The sBSANPs were characterized by a small size of about 134–264 nm, a negative charge of −37 to −40 mV, an entrapment efficiency of 59–71%, and a particle yield of 65–86%. The cytotoxicity assays using sBSANPs showed a higher potency specifically against both MDA-MB-468 and MDA-MB-231 cells (ER−, PR−, HER2−) compared to MCF-7 (ER+, PR+, HER2−), and exhibited an extremely low IC_50_ at the nanomolar levels (2.01–6.82 nM). The release profiles indicated that SN-38 presented an initial burst release within 12 h, and sBSANPs had a slow release pattern. Flow cytometry results showed that the fluorescence intensity of sBSANPs was significantly higher than that of the control group. The confocal images also confirmed that sBSANPs were taken up by MDA-MB-468 cells. Moreover, we found that a larger BSANP size resulted in an increased hemolytic effect. In vivo animal studies demonstrated that loading of SN-38 into bovine serum albumin nanoparticles could minimize the initial concentration without extending the elimination half-life, but significantly minimized the Cmax (*p* < 0.001) as compared with irinotecan treatment.

## 1. Introduction 

Triple-negative breast cancer (TNBC), by definition, refers to all breast cancers characterized by a lack estrogen, progesterone, and *HER2* expression. Individuals with TNBC have been found to have worse overall survival (OS) when compared with individuals with ER+/HER2−breast cancers, and the difference was found to be most pronounced in the first two years by a large observational prospective study [1]. Until now, chemotherapy-based regimens in adjuvant or neoadjuvant settings, specifically anthracyclines, have been the choice of treatment prior to the advent of targeted therapy. Nab-paclitaxel, also known as Abraxane^®^, is a product consisting of nanoparticles around 130 nm in size, and is indicated for metastatic breast cancer after the failure of combination chemotherapy for metastatic disease, which is common in the case of TNBC [2]. Recently, the combination of nab-paclitaxel and PD-L1 has resulted in a breakthrough in TNBC treatment by improving both progression-free and overall survival [3]. In this regard, albumin-based formulations continue to show their value and advance TNBC therapy. 

Albumins are potential biomimetic materials that are known not only to carry both hydrophilic and hydrophobic molecules, but have been reported to have binding sites for camptothecins on subdomain IB [4]. In addition, albumin nanoparticles have been suggested to be taken up though GP60/SPARC receptors, which are overexpressed in colon, prostate, pancreatic, and breast cancers, thus leading to an intensified anticancer effect. 

In February 2016, an antibody-to-SN-38 conjugate, IMMU-132, developed by Immunomedics, received the “breakthrough therapy” designation from the US Food and Drug Administration (FDA) for the treatment of patients with TNBC. SN-38 (7-ethyl-10-hydroxycamptothecin), a topoisomerase I inhibitor, is an active metabolite of irinotecan. Although irinotecan is extensively used clinically, only 2–8% of irinotecan is converted into SN-38 in vivo [5], and a higher dosage of irinotecan is thus required to achieve the desired therapeutic effect [6,7]. In this regard, replacing irinotecan directly with SN-38 might be an option for improving therapy.

In vitro, SN-38 demonstrates 100- to 1000-fold greater potency than irinotecan [8]. Although this high potency of SN-38 against cancer cells is desirable, the shift between lactone and carboxylate forms of SN-38, in addition to causing life-threatening delayed diarrhea, impedes clinical application of SN-38. Additionally, SN-38 is a biopharmaceutical classification system (BCS) IV drug, which is the least desired classification for drug development [9]. Delivery strategies of SN-38 can usually be categorized into two major methods: physical and chemical. Physical methods include nanoparticles such as liposomes, micelles, and cyclodextrin. Chemical methods include poly(lactic-*co*-glycolic acid)(PLGA) polymer/peptide/albumin/or antibody–drug conjugates such as IMMU-132 [10]. Recently, several advance delivery strategies have been demonstrated, including encapsulation methods and conjugation or PEGylating methods for various cancer applications, such as linoleic acid-SN38 (LA-SN38) conjugate polymeric nanoparticle, chitosan-coated PLGA nanoparticles, and macrophages as an active tumor-targeting carrier.

The aim of this study was to investigate the applicability of SN-38-loaded albumin nanoparticles in the treatment of TNBC. We optimized the SN-38-loaded albumin nanoparticles and characterized their physical properties and in vitro release pattern. Effects were assessed on both the MCF-7 and MDA-MB-468 cell lines using an in vitro cytotoxicity assay. Assays for safety included a culture test and hemolysis assay. In vivo animal pharmacokinetics was examined to understand the effects on circulation.

## 2. Materials and Methods

### 2.1. Materials

Bovine serum albumin (heat shock fraction, pH 7, ≥98%) and rhodamine 6G (Rho) were purchased from Sigma-Aldrich (St. Louis, MO, USA). The glutaraldehyde solution (25% for synthesis) used for crosslinking was purchased from AppliChem GmbH (Darmstadt, German) and 1-ethyl-3-(3-dimethylaminopropyl)-carbodiimide hydrochloride was from Tokyo Chemical Industry (Tokyo, Japan). Dulbecco’s modified Eagle’s medium (DMEM) and fetal bovine serum were from Gibco (New York, NY, USA). The Hoechst 33,342 solution and apoptosis kit were from Invitrogen (Waltham, MA, USA). The Cell Counting Kit-8 was from Enzo Life Sciences (New York, NY, USA). Ultrapure water (Merck, Darmstadt, Germany) was used for all experiments. All other reagents used were of analytical grade.

### 2.2. Preparation of sBSANP, BSANP, and Rho-sBSANP

SN-38-entrapped albumin nanoparticles (sBSANP) were prepared by the desolvation method. Three formulations corresponding to different amounts of bovine serum albumin (BSA) were prepared. sBSANP-F75, -F62.5, and -F40 solutions were prepared using, respectively, 75, 62.5, and 40 mg of BSA lyophilized powder dissolved in 1 mL ultrapure water. The BSA was allowed to fully dissolve before adding 0.5 mg of SN-38. The solution was then sonicated with a probe-type sonicator, on ice, at 30 W amplitude until no undispersed SN-38 could be observed, which took 5 min. Anhydrous ethanol (2 mL) was added at a flow rate 0.5 mL/min into the sonicated solution under stirring at 550 rpm. Next, 44.2, 36.8, and 23.6 µL of 8% glutaraldehyde was respectively added dropwise, and stirred overnight. The crosslinked formulations were purified using three cycles of centrifugation and redispersion, and finally redispersed to 1 mL in PBS (pH 7.4). 

The Rho-sBSANPs were prepared by post-modification [11]. The prepared sBSANP-F75, -F62.5, and -F40 were activated by 1% 1-ethyl-3-(3-dimethylaminopropyl) carbodiimide (EDC) dissolved in phosphate buffer (pH 8) with volumes of 75, 62.5, and 40 µL, respectively. The mixtures were incubated for 15 min with constant shaking at room temperature before 37.5, 31.3, and 20.0 µL of the freshly prepared rhodamine 6G solution (1 mg/mL) were added to the formulations, respectively. The mixtures were then incubated again for 3 h with constant shaking at room temperature. The purification procedure was the same as that used in the sBSANP preparation.

### 2.3. Characterization of the sBSANPs

#### 2.3.1. Size, Polydispersity Index, Zeta Potential

The particle size and zeta potential were analyzed by dynamic light scattering (DLS) (ELSZ-2000, Otsuka Electronic, Hirakata, Japan) using samples diluted 1:20 with ultrapure water.

#### 2.3.2. Particle Yield

The particle yield was calculated according to the Equation (1)
(1)$$particles\;yield\;\%=\;\frac{W_{1}-W_{0}\;}{W_{theoretical}\;}\times 100\;\%,$$
where $W_{0}$ is the weight of each empty Eppendorf tube; $W_{1}$ is the total weight of the dried pellet and Eppendorf tube; and $W_{theoretical}$ is the sum of BSA and SN-38 in the formulation. In brief, each empty Eppendorf tube was weighed before 1 mL of formulation was placed into the Eppendorf. The formulation was then centrifuged, and the supernatant was discarded. The pellet in the open Eppendorf tube was kept under vacuum overnight. After the pellet was thoroughly dried, the pellet along with the Eppendorf was weighed again to calculate the particle yield.

#### 2.3.3. Entrapment Efficiency Calculation

The entrapment efficiency of sBSANPs was calculated as Equation (2): (2)$$\mathrm{EE}\;\%=\;\frac{C_{total}-C_{free}}{C_{total}},$$
where $C_{total}$ is the amount of drug placed in the formulation, and $C_{free}$ is the supernatant collected after centrifugation. The supernatant was further centrifuged at 80,000 RCF for 30 min to eliminate the potential remaining nanoparticles. To concentrate SN-38, 400 µL of the centrifuged supernatant was dried overnight, followed by redissolving in 100 µL Dimethyl sulfoxide (DMSO_ and 100 µL PBS (pH 7.4) for HPLC analysis.

### 2.4. Fluorescence Quenching Study

The preparation of BSA solutions with SN-38 for the fluorescence quenching study was identical to the preparation of sBSANP. The fluorescence spectrophotometer was a model F-4500 (Hitachi, Tokyo, Japan). The excitation wavelength was 280 nm (slit = 2.5 nm) and the emission wavelength was 300–700 nm (slit = 10 nm). The samples were kept in dry bath incubators at 25, 30, and 37 °C prior to measurement. Fluorescence quenching can be described using the Stern–Volmer Equation (3) [12]:(3)$$\frac{I_{0}}{I}=1+\;K_{sv}\left[Q\right],$$
where *I*_0_ is the fluorescence intensity of the BSA alone; *I* is the intensity of BSA in the presence of SN-38; *K_sv_* is the Stern–Volmer constant; and [*Q*] is the concentration of SN-38.

### 2.5. FTIR

The samples were diluted with approximately 1% KBr mixing powder and pressed into 16 mm disks at 15 t for analysis using a Fourier-transform infrared (FTIR) spectrometer (Vertex 70 v, Bruker, Billerica, MA, USA). The wavenumbers ranged from 4000–380 cm^−1^ at a resolution of 1 cm^−1^.

### 2.6. In Vitro Release Kinetics 

A total of 1 mL of the sample was placed in dialysis tubing (MWCO = 6000–8000 kDa). Each dialysis bag was fully submerged in 5 mL receptor medium, which comprised PBS with 99% ethanol (1:1 (*v*/*v*)) at pH 7.4, and stirred at 100 rpm at a constant temperature of 37 °C. The receptor medium was sampled at 0.5, 1, 2, 4, 6, 8, 10, 12, 24, 48, and 72 h and analyzed by HPLC. The samples were immediately supplemented with an equal volume of fresh receptor medium after each sampling. The samples in this study included sBSANP-F75, -F62.5, and -F40, and the SN-38 solution. The solvent for SN-38 was the receptor medium and the concentration was the theoretical concentration of sBSANPs (0.5 mg/mL). Release profiles and correlation coefficients of SN38 free solution and SN38-loaded sBSANPs were investigated, as shown in Appendix A. The cumulative release percentage data were plotted and fitted based on five mathematical models: zero-order, first-order, Higuchi, Korsmeyer–Peppas, and Hixson–Crowell.

### 2.7. Ex Vivo Hemolysis Assay

Red blood cells (RBCs) were collected from male adult Sprague Dawley rats. Collected blood was centrifuged at 2000 RCF for 10 min, followed by re-dispersal three times in PBS (pH 7.4). Thereafter, the RBCs were dispersed in PBS (pH 7.4) (1:9 (*v*/*v*)) as the stock solution and stored at 4 °C for later use. To analyze hemolysis, 100 µL stock solutions were mixed with 50 µL sBSANP or BSANP formulations along with 2.45 mL PBS (pH 7.4). The mixture was incubated at 37 °C under constant shaking for 1 h, and centrifuged at 2000 RCF for 5 min to separate the hemoglobin from cell debris. Finally, the supernatant was assayed at 415 nm using a UV spectrometer. The measured value for the positive control was RBCs dissolved in water and was taken as 100% hemolysis, while the value for the negative control was RBCs dissolved in PBS (pH 7.4) and was taken as 0% hemolysis [13].

### 2.8. Cell Culture and Cytotoxicity Assay

#### 2.8.1. Cell Culture

The cell lines MDA-MB-468 and MCF-7 were purchased from American Type Culture Collection (ATCC, Manassas, VA, USA); HUVEC and Hs68 were from the Bioresource Collection and Research Center (BCRC, Hsinchu, Taiwan). The culture medium was DMEM containing 10% fetal bovine serum and 1% liquid penicillin–streptomycin. The medium was changed twice a week. The cells were kept in a 5% CO_2_ incubator at 37 °C.

#### 2.8.2. Cytotoxicity Assay

The cells lines, including MDA-MB-468, MDA-MB-231, MCF-7, HUVEC and Hs68, were transferred to 96 well plates and incubated overnight to allow the cells to attach (in logarithmic phase). The culture medium was then removed and replaced with the SN-38 DMSO solution, sBSANP, and BSANP diluted in culture medium, and incubated for 96 h. After 96 h, CCK-8 was added, followed by incubation in a CO_2_ incubator for 3 h. The absorbance was measured using a microplate spectrophotometer at 450 nm. The cell viability was calculated by Equation (4):(4)$$\mathrm{Cell}\;\mathrm{viability}\;\%=\;\frac{Ab_{sample}-Ab_{blank}}{Ab_{control}-Ab_{blank}}\times 100\;\%,$$
where Ab denotes the absorbance.$\;Ab_{control}$ represents 100% survival and $Ab_{blank}$ represents no cells. 

### 2.9. Cellular Uptake

#### 2.9.1. Flow Cytometry

Cellular uptake of the nanoparticles into the MDA-MB-468 cells was evaluated by flow cytometry. Prior to treatment, the cells were transferred to a 6 well plate and incubated overnight. After incubation, Rho-sBSANP formulations at 100, 10, and 1 nM were added, followed by incubation for 4 h. The cells grown in the absence of the nanoparticles were used as the control. Flow cytometry was performed on a Cytomics FC500 (Beckman Coulter, Brea, CA, USA). The wavelength of flow cytometry was 488 nm/575 nm (E_x_/E_m_). 

#### 2.9.2. Confocal Laser Scanning Microscopy

Cellular uptake was further inspected by confocal laser scanning microscopy. MDA-MB-468 was transferred to a 6 well plate in which a glass coverslip had been placed prior to transfer. After overnight incubation, the culture medium was replaced by 100 nM Rho-sBSANP formulations in culture medium for 24 h. At the designated timepoint, medium was removed and replaced with a diluted Hoechst 33342 solution (Invitrogen, Waltham, MA, USA) to stain nuclei. Next, the cells on the glass coverslips were washed with PBS (pH 7.4). Finally, the glass coverslip was carefully placed onto a slide and sealed with nail polish to prevent drying. The finished slides were stored at 4 °C until use.

### 2.10. Annexin V–PI Apoptosis Assay

Cells were transferred to a 6 well plate and incubated overnight. After incubation, the culture medium was replaced with 3 mL of the diluted sBSANP formulation at 100 nM, and incubated for 4, 24, 48, and 96 h. At the designated time, the cells were processed according using Apoptosis Kit V13241 (Invitrogen, Waltham, MA, USA), and then analyzed on a Cytomics FC500 (Beckman Coulter, Brea, CA, USA).

### 2.11. Pharmacokinetic Study

The animals used the in vivo evaluation were 12 week old male Sprague Dawley rats from BioLASCO Taiwan Co. Ltd. All animal experiments were conducted in accordance with institutional guidelines and approved by the Animal Care and Use Committee of Kaohsiung Medical University, Kaohsiung, Taiwan. (NO. IACUC-105241 Date 28 October 2016) All animals were starved overnight prior to experiments.

The samples were administered through a femoral vein catheter. The surgical procedure was performed according to Jespersen et al. (2012) [14]. Before femoral vein catheter surgery, the rats were anesthetized using a combination of (1) Zoletil^®^ at 20 mg/kg (tiletamine hydrochloride 10 mg/kg and zolazepam hydrochloride 10 mg/kg) and (2) Ropum^®^ 10 mg/kg (xylazine hydrochloride). Under this dosage, the rats were routinely anesthetized for 5–6 h. The rats were administered with 1 mL sample at a rate of 0.1 mL/min. This dosage is generally considered to be good practice for intraveniousinjection (< 5 mL/kg) [15,16]. The samples were sBSANP-F62.5 or equivalent SN-38 in the vehicle (315 µg/mL), where the formula of the vehicle solution was 10% DMSO, 20% Cremophor EL, and 70% PBS (pH 7.4) [17]. The blood was sampled through the same femoral catheter at 20, 40, 60, 90, 120, 180, 240, and 300 min. 

The correlation coefficient for the pharmacokinetic compartmental model was calculated and data predicted using Phoenix WinNonlin V8.1 software (Certara, St. Louis, MO, USA). The area under the concentration–time curve (AUC_0–∞_) was calculated by the trapezoidal rule. The half-life (t_½_) and clearance (Cl) values volume of distribution (Vd) were also obtained. The Cl value was calculated as the dose/AUC_0–∞_, and area under the first moment curve (AUMC) and mean residence time (MRT) were obtained by summation of the central and tissue compartments.

### 2.12. High Performance Liquid Chromatography (HPLC) Analysis

#### 2.12.1. In Vitro Quantitation 

In vitro qualitative analysis was performed using a HPLC-UV 5000 series (Hitachi, Tokyo, Japan) with Purospher STAR RP-C18 (250 × 4.6 mm, 5 µm) column at 30 °C. The assayed wavelength was 265 nm. The mobile phase was composed of 75 mM ammonium acetate (pH 6.4) and acetonitrile (ACN) in a gradient fashion, where the initial ACN concentration was 20%, which was increased to 65% at 13 min and back to 20% at 14 min, before the conclusion of the experiment at 15 min. The retention time for the SN-38 carboxylate form was 6.06 min and 9.55 min for the lactone form.

#### 2.12.2. In Vivo Quantitation

For the in vivo qualitative analysis, the HPLC-FL L-2000 series (Hitachi, Tokyo, Japan) was used at λ_ex_ = 368 nm, λ_em_ = 515 nm with an encapped Discovery HS C18 (150 × 4.6 mm, 5 µm) column at 30 °C [18,19]. The mobile phase was identical to that of the in vitro analysis, except that the duration was extended from 15 min to 19 min to ensure no flatulence before the next injection. The retention time of SN-38 was 8.17 min corresponding to a single peak.

### 2.13. Statistical Analysis

Statistical analyses were performed by SigmaPlot 12.0 using the *t*-test, one-way or two-way analysis of variance (ANOVA). ANOVA. A 0.05 level of probability was taken as the level of significance. Asterisks (***) denote *p* < 0.001. All data are expressed as the mean ± standard deviation (SD).

## 3. Results

### 3.1. Attributes Affecting the Size of Albumin Nanoparticles

To find the optimal conditions for sBSANPs between 130 and 250 nm, a series of pre-experiments were conducted in the following order: albumin concentration, pH value of albumin solution, ethanol volume, and ethanol rate. Albumin nanoparticles without SN-38 (BSANP) were used to optimize these conditions.

Figure 1 shows the relationships between particle size and the four attributes. Figure 1A shows that a range of 40–75 mg/mL corresponded to smaller particle sizes, and was adopted for the preparation of sBSANPs. Figure 1B shows that the size was the smallest (172.97 ± 20.16 nm) at pH 7.5, and increased as the pH value decreased. Precipitation was observed at pH 6.0, 5.0, and 4.5, and the size was not measured. For the ethanol volume, the volume ratio of ethanol and albumin solution was positively correlated to particle size and, thus, the volume ratio of 2 was selected. Regarding the ethanol rate, 0.5 mL/min corresponded to the smallest size (180.73 ± 8.06 nm). Figure 1C shows that the BSANP size increased when the ethanol/albumin ratio increased. Figure 1D shows that the desolvation rate did not drastically affect the BSANP size, compared with the aforementioned size-influencing attributes, as the sizes only ranged from 180 to 230 nm. Still, 0.5 mL/min was chosen, as this rate corresponded to the smallest particle size.

Overall, three different albumin concentrations of 75, 62.5, and 40 mg/mL were selected for sBSANP preparation with a pH ~7 (unadjusted albumin solution), ethanol/albumin solution volume ratio of 2, and an ethanol rate of 0.5 mL/min.

### 3.2. Characterization of sBSANPs 

#### 3.2.1. Particle Size, Zeta Potential, Entrapment Efficiency, and Particle Yield

Table 1 shows the characteristics of the sBSANPs, BSANPs, and Rho-BSANPs. Overall, the particle size, entrapment efficiency, and particle yields were proportional to the amount of albumin in each formulation. Appendix A shows the SEM image of the sBSANP and SN-38 crystals.

#### 3.2.2. Fluorescence Quenching Study

We attempted to investigate whether the probe-sonication process not only dispersed the SN-38 crystal, but also increased the affinity of SN-38 to albumin molecules. Figure 2A shows the results of different albumin concentration at 75, 62.5, and 40 mg/mL mixed with SN-38 without probe sonication. The fluorescence intensity of albumin was quenched in response to increasing SN-38, indicating that the SN-38 molecules interacted with albumin. 

The results of the albumin solution mixed with SN-38 showed that the weights of both were equivalent to sBSANP-F75, -F62.5, and -F40. Here, we demonstrated the fluorescence intensity of sBSANP-F75, -F62.5, and -F40 and the BSA-only solution with and without probe sonication. Probe sonication did not alter the fluorescence intensity of the BSA-only solution. However, in the presence of SN-38, probe sonication intensified the fluorescence quenching effect, suggesting that probe sonication increased the affinity between SN-38 and albumin. Furthermore, the mechanism of fluorescence quenching before and after probe sonication was studied. Figure 2B shows that the Stern–Volmer plot switched from linear (before probe sonication) to cursive (after probe sonication), suggesting that two quenching mechanisms were involved after probe sonication, thus strengthening the affinity between SN-38 and albumin molecules.

#### 3.2.3. FTIR

Figure 3 shows the spectra of SN-38, sBSANP, BSANP, and BSA lyophilized powder. The spectra of sBSANP were identical to that of the BSA lyophilized powder, implying that no additional functional group had formed. Furthermore, the spectra of sBSANP shared no peaks with SN-38, indicating that unentrapped SN-38 molecules were eradicated during the three cycles of purification during sBSANP preparation.

### 3.3. In Vitro Release Kinetics 

The release kinetics of all three sBSANP formulations reached a plateau at 12 h, and demonstrated linearity from 0 to 12 h (Figure 4). During this period, the release pattern of all three formulations fit the zero-order and Hixson–Crowell models best (Appendix A).

### 3.4. Ex Vivo Hemolysis Assay

In Figure 5, the BSA solution concentrations corresponding to the three formulations did not exhibit a hemolytic effect. By contrast, both BSANPs and sBSANPs of F75, F62.5, and F40 exhibited a hemolytic effect ranging from 16.98 ± 0.44 to 4.21 ± 0.70. The hemolytic effects were positively correlated to the albumin amount in each formulation—in other words, they correlated to the particle size. The hemolytic effect did not show any differences between the BSANPs and sBSANPs in each formulation.

### 3.5. In Vitro Cytotoxicity and Safety Evaluation

Figure 6 shows the cytotoxicity assay for the MDA-MB-468, MDA-MB-231, MCF-7, and HS68 cell lines. As seen in Table 2, the IC_50_ values of MDA-MB-468, MDA-MB-231, and MCF-7 treated with sBSANPs matched the IC_50_ values seen with the use of SN-38 solution. None of the tested cell lines were responsive to the BSANP formulations. 

### 3.6. Cellular Uptake

Figure 7A,B suggests that the fluorescence was measurable in MDA-MB-468. The results for Rho-sBSANP-F75 showed the lowest fluorescence intensity, while Rho-sBSANP-F62.5 showed the highest.

Figure 8 shows that the Rho-sBSANP formulations were observable in the cytoplasmic region of MDA-MB-468 under CLSM at 600× magnification.

### 3.7. Annexin V–PI Apoptosis Assay

Evidence of apoptosis is shown in Figure 9. At 48 h, the MDA-MB-468 cells treated with sBSANP formulations or the SN-38 solution started to appear in the lower quadrant (the early apoptosis region). At 96 h, the cells could be observed in all four quadrants, including late apoptosis and necrosis, corresponding to the upper right and upper left regions, respectively.

### 3.8. Pharmacokinetic Study

Figure 10 shows the SN-38 plasma concentration when administering sBSANP-F62.5, or SN-38 in vehicle as a comparison. Both plasma profiles fit the two-compartment model best (Rsq > 0.909); however, the pharmacokinetic parameters estimated with the two-compartment model did not show any significance regarding either clearance or elimination half-life (Table 3). 

## 4. Discussion

### 4.1. Attributes Affecting the Size of Albumin Nanoparticles

The preparation of albumin nanoparticles by the desolvation method involves the consideration of many attributes, such as albumin concentration, pH of the albumin solution before desolvation, the desolvating agent type and amount, the desolvation rate, the crosslinking agent type and amount, the temperature during crosslinking, stirring rate, etc. The desolvating agent type, crosslinking agent type and amount, and stirring rate used in this study were based on previous reports [11,20]. 

Regarding the albumin solution concentration, the choice of concentration can vary greatly. The solubility of BSA is 40 mg/mL [21]. To the best of our knowledge, unsaturated concentrations have been applied in some studies. For example, 3 mg/mL was used by Li et al. (2012) [22], and 1 mg/mL was used by Ruttala et al. (2015) [23]. By contrast, most studies have selected the use of a supersaturated concentration (above 40 mg/mL). For example, 50 mg/mL was used by Galisteo–González et al. (2014) [24], 100 mg/mL by Kouchakzadeh et al. (2014), and 250 mg/mL by Yang et al. (2007) [25]. Therefore, considering that the majority of studies chose to use a supersaturated concentration and that a higher amount of albumin presumably results in a higher entrapment efficiency, we chose quantities of 30, 40, 50, 62.5, 75, and 100 mg/mL to obtain the desired sizes in the pre-experiment. The results demonstrated that a range of 40–75 mg/mL corresponded to smaller particle sizes, and was adopted for the preparation of sBSANPs. 

Regarding pH, we adjusted the pH of the albumin solution using phosphoric acid or NaOH solution to assay its effects on size. Figure 1B shows that the size was the smallest (172.97 ± 20.16 nm) at pH 7.5, and increased as the pH value decreased. Precipitation was observed at pH 6.0, 5.0, and 4.5, which was in agreement with the results of Kouchakzadeha et al. (2014) [20]. The precipitation can be explained by the isoelectric point. As the pH value approaches the pI (albumin pI = 4.7 [21]), the albumin molecules aggregate as the net charge is close to neutral. Here, considering that the active form of SN-38 dominated at pH < 6 [9,26], which was an infeasible condition for BSANP formation, we decided to leave the pH value of the albumin solution unadjusted (pH ~7), although this was not the most ideal condition when considering nanoparticle size.

Regarding the amount of the desolvation agent, most previous studies have used an ethanol/albumin solution ratio between 2 and 4 [11,20,24,27]. We conducted a series of experiments using different ratios from 2- to 10-fold. Figure 1C shows that the BSANP size increased when the ratio increased. Therefore, we chose the smallest ratio for sBSANP preparation, which was 2 mL of ethanol to 1 mL of albumin solution.

Finally, different desolvation rates were also investigated from 0.5 to 5 mL/min. Li et al. (2008) reported a significant increase in size due to a higher ethanol addition rate [28]; in the study by Galisteo–González et al. (2014), by contrast, the size decreased as the addition rate increased [24]. This inconsistency in the literature made the investigation of the ethanol addition rate indispensable. 

### 4.2. Characterization of sBSANP

Due to the low solubility of SN-38, SN-38 is impossible to dissolve in aqueous solutions, even when applying water bath sonication. Therefore, a probe-type sonicator was applied to assist with SN-38 crystal dissolution. This technique is based on that used by Kouchakzadeh et al. (2014) for the preparation of tamoxifen albumin nanoparticles. After probe sonication, SN-38 became evenly distributed in the albumin solution.

The size of these formulations demonstrated a good response to different amounts of albumin (Table 1). sBSANP-F40 could be an ideal choice, as the observed particle size was close to 130 nm, like Abraxane^®^, though the particle yield rate was poor (65.91 ± 0.78%) and the entrapment efficiency (59.14 ± 3.14%) was the lowest among the three formulations, although not significantly different from that of sBSANP-F62.5. The observed PDI (polydispersity) values were below 0.3, suggesting that the preparation of albumin nanoparticles using the aforementioned parameters was capable of yielding monodispersive particles [29]. These results were supported by SEM, which showed that the sBSANP formulations were monodispersive (Appendix A). 

Fluorescence quenching is a common technique used to study the interaction between a protein and a drug (the quencher) [30]. Although no SN-38 crystal was observed visually, we conducted a fluorescence quenching experiment to study the interaction between albumin and SN-38. The fluorescence of BSA depends on three fluorophores: tryptophan (Trp), tyrosine (Tyr), and phenylalanine (Phe). Due to the low fluorescence intensity (low quantum yield) of the Phe residue, the fluorescence intensity of BSA can be primarily attributed to Trp and Tyr residues. The optimal excitation wavelength for both Trp and Tyr is 280 nm, with $\lambda_{em}^{max}$ at approximately 365 nm [31,32]. 

The quenching capability of SN-38 was confirmed by using BSA solution in the presence of various concentrations of SN-38 at 298 K (Figure 2A). The fluorescence intensities of the solutions (BSA only) before and after probe sonication were identical, which indicates that probe sonication did not affect the fluorescence of albumin. When SN-38 was added to the BSA solution, however, the intensity decreased and then further decreased after probe sonication. Briefly, the results suggested that SN-38 does serve as a fluorescence quencher, and that probe sonication further assisted SN-38 in coming close to the Trp and Tyr residues inside the BSA molecule.

Quenching mechanisms are usually classified as dynamic, static, or both. Dynamic quenching, also known as collisional quenching, is defined as when the quencher assists the fluorophore in returning to the ground state without proton emissions upon contact with the fluorophore; since the collision of the quencher to fluorophore depends on diffusion, the *Ksv* increases as the temperature increases. Static quenching is defined as the formation of a non-fluorescent complex between the quencher and fluorophore; since the higher temperature results in an unstable quencher–fluorophore complex, the *Ksv* decreases as the temperature increases [33]. On one hand, these two mechanisms displayed a linear regression in the Stern–Volmer plot. On the other hand, the mix of dynamic and static quenching resulted in an upward curvature [31], since there is a [*Q*]^2^ term in the Stern–Volmer Equation (5):(5)$$\frac{I_{0}}{I}=\left(1+\;K_{sv}\left[Q\right]\right)\left(1+\;K_{a}\left[Q\right]\right),$$
where *K_a_* denotes the association constant of the quencher–fluorophore complex [34]. The other terms were the same as those aforementioned.

Figure 2B shows the Stern–Volmer plot of the BSA solution in the presence of an increasing concentration of SN-38 before probe sonication at different temperatures. The plot demonstrated high linearity, and the *K_sv_* value increased with temperature, indicating dynamic quenching. The results also showed that the BSA solution with SN-38 after probe sonication had a combination of dynamic and static quenching because the plot showed upward curves, and the *K_sv_* value decreased as the temperature rose. We deduced that the probe sonication not only dissolved the SN-38 crystals, but facilitated the formation of the quencher–fluorophore complex. In other words, probe sonication might have enhanced the drug–albumin affinity.

FTIR is a useful tool for evaluation of the surface chemistry of particles after modification and the change in the secondary structure of proteins [23,35]. Figure 3 shows that the sBSANP spectrum was identical to the BSANP and BSA lyophilized powder in the fingerprint region, and shared no single peak with SN-38. The former implies that the albumin molecule of sBSANP or BSANP still retained its original condition, although new bonds should have emerged due to crosslinking; the latter implies that no SN-38 was detected on the surface of the sBSANPs and all unentrapped SN-38 molecules were thoroughly removed [23], thus perhaps preventing false positive results in the following in vitro/in vivo evaluations.

Additionally, the amide I/II ratio can be used to evaluate the secondary structure of proteins during denaturation. This ratio increases as the protein undergoes denaturation, and decreases as the peptides become increasingly ordered structures (beta sheets or alpha helices). Here, the result was as expected: the ratio of the nanoparticles was higher than that of the BSA lyophilized powder (amide II/amide I–BSA powder = 0.70; BSANP-F62.5 = 0.67; sBSANP-F62.5 = 0.65).

### 4.3. In Vitro Release Kinetics 

Figure 4 demonstrates the release kinetics of all three sBSANP formulations and control. The control was SN-38 in receptor medium, of which the concentration was equivalent to the theoretical concentration of sBSANPs (0.5 mg/mL). The three sBSANP formulations did not exhibit any differences in the release model or release amount. 

The release models of the three formulations fit the zero-order and Hixson–Crowell model best among the five different mathematical models (Appendix A). The good fit to the zero-order of sBSANP corresponds to a desirable release profile, because it suggests that the drug release was independent of concentration and occurred at a constant rate, thus leading to better control of the concentration in clinical applications [36]. On the other hand, the Hixson–Crowell model could explain why the three formulations were so close in terms of the release kinetics. The Hixson–Crowell model describes how the release occurs according to the surface area of the particles, which is proportional to the cube root of its volume. Here, although the volume (particle size) ratios of F75/F62.5 and F62.5/40 were 1.20 and 1.56, respectively, the drug released, according to the Hixson–Crowell model, could be further decreased to $\sqrt[{3}]{75}/\sqrt[{3}]{62.5}$ and $\sqrt[{3}]{62.5}/\sqrt[{3}]{40}$, which were 1.07 and 1.14, and explained the similarity in the release kinetics.

The fit of the release models was calculated using the data from 0 to 12 h, because the results showed good linearity within this range, and lost linearity after 12 h. This loss of linearity was expected, since it has been found in every reported release test using dialysis membranes. This trend was also observed for the control. Therefore, we did not consider that the loss of linearity was due to the formulation design, and should instead be viewed as a limitation of the dialysis membrane.

### 4.4. Ex Vivo Hemolysis Assay

The hemolysis assay is a useful tool for conducting a preliminary evaluation of the toxicity of pharmaceutical excipients. Exposure to foreign objects causes hemolysis. Due to genetic variation, some individuals are more vulnerable to hemolysis, even at low concentrations. Having received growing attention, the hemolysis assay has also been recommended by the FDA in evaluating the hemolytic potential of IV route materials in the *Guidance for Industry Nonclinical Studies for the Safety Evaluation of Pharmaceutical Excipients* [37]. 

The hemolysis–nanoparticle relationship has been reported in several studies [38,39]. The hemolytic effect may be caused either by the oxidative stress of generated reactive oxygen species (ROS), or simply through mechanical damage of the cell membrane, and can be size-dependent or concentration-dependent [40]

Figure 5 shows a size-dependent hemolytic effect where the bigger the size, the more severe the hemolysis was. Even though it has been reported that albumin inhibits nanoparticle-induced hemolysis at a very low concentration [40], this may be limited to native albumin and may not apply to crosslinked albumin nanoparticles. Regarding the hemolytic effect of SN-38, all three sBSANP formulations were not statistically different from the BSANPs in each group, supporting the FTIR result that showed that the unentrapped SN-38 molecules had been removed completely, and therefore did not contribute to any additional hemolytic effects.

### 4.5. In Vitro Cytotoxicity and Safety Evaluation

Three cell lines were used in this study. The cell lines MDA-MB-468 and MDA-MB-231 are basalA and B-like triple-negative breast cancer cell lines, and the basal-like subtype accounts for 70% of TNBC cases [41], whereas MCF-7 is a human estrogen positive (ER+) and progesterone positive (PR+) cell line that is extensively used to evaluate hormone-based therapies. HUVEC is human umbilical vein endothelial cell and Hs68 is a health human fibroblast cell lines, which have been applied to evaluate oxidative stress to assess safety [42]. 

In Table 2, the IC_50_ values in MDA-MB-231 and MDA-MB-468 were 5- to10-fold lower than that for MCF-7. This suggests that MDA-MB-468 (TNBC) was more sensitive to SN-38 cytotoxicity than MCF-7 (non-TNBC). The IC_50_ values of the three sBSANP formulations matched the IC_50_ of SN-38 solution in the MDA-MB-231, MDA-MB-468, and MCF-7 cell lines, suggesting that the preparation of sBSANPs did not alter its anticancer effect toward cancer cell lines. These IC_50_ values were also in good agreement with previously reported results by Goldenberg et al. (2015) [43].

The BSANP formulation was regarded as showing no toxicity to all cell lines, because the IC_50_ plots neither demonstrated sigmoidal curves (Figure 6) nor passed the 50% survival rate.

Past research has also mentioned that SN-38-loaded PLGA nanoparticles could act against highly aggressive breast cancer metastasis in 4T1 mouse mammary breast cancer [44]. It has been suggested that SN-38 could have a higher potency against metastasized breast cancer. However, the safety issue has yet to be resolved. 

Regarding safety evaluation, Hs68 and HUVEC cells was found to be surprisingly insensitive to SN-38 at the same concentrations; again, the curve neither showed sigmoidal curves nor passed the 50% rate. 

After confirming that the sBSANPs showed toxicity toward MDA-MB-468, we continued to study the mechanism of the cytotoxicity effect, regardless of whether it came from the gradual release of SN-38 from the albumin nanoparticles or whether these particles were actually taken up by the cells.

Figure 7A shows that the fluorescence intensities of the Rho-sBSANP formulations were significantly higher that of the control, which confirmed that these particles could be taken up by MDA-MB-468. Therefore, we deduced that the cytotoxicity mechanism of sBSANP came from direct uptake, instead of the gradual release of SN-38 from nanoparticles. If this mechanism is still applicable in vivo, we might expect less adverse effects due to the enhanced permeability and retention (EPR) effect, as nanoparticles should be inside cancer cells rather than remaining in the circulatory system.

In Figure 7B, a specific size effect was found. F62.5 at 100/10 nM showed the highest intensity. Before this study, we expected that either Rho-sBSANP-F40 or F75 would have the highest intensity. The assumption that F40 would be highest was based on the understanding that the smaller the particle, the better the cellular uptake. By contrast, with F75 particles being the largest among the three formulations, they would entrap the largest amount of rhodamine 6G, which would result in the highest fluorescence intensity. Although F62.5 resulted in the best cellular uptake, further study is necessary to corroborate the expectation.

The confocal image supports the observation that the Rho-sBSANP formulations were uptaken by MDA-MB-468. Figure 8 reveals that the cytosol was filled with Rho-sBSANPs.

Identification of the mechanisms of cytotoxicity, apoptosis, or necrosis is essential. Apoptosis is a form of carefully regulated cell death; by contrast, necrosis is caused by physical stress or toxin exposure, such as membrane damage caused by temperature or oxidative stress by hydrogen peroxide. Necrosis is undesirable as it is involves inflammation and can be fatal for living organisms [45].

Figure 9 shows the annexin V–PI results in a time scale from 4 to 96 h. From 4 to 48 h, an increase in percentage was observed in the lower right quadrant, indicating the activation of early apoptosis. At 96 h, late apoptosis along with necrosis was observed. The total of 39.72% in the necrosis section of F40 was higher than that of the others, which is consistent with the cytotoxicity results. By contrast, 49.98% in the apoptotic section of F62.5 was the highest among the three sBSANP formulations, which supports the flow cytometry results where Rho-sBSANP-F62.5 had the highest fluorescence intensity in MDA-MB-468.

### 4.6. Pharmacokinetic Study

The sBSANP-F62.5 was selected for further in vivo study. From the aforementioned in vitro experiments, the three sBSANP formulations did not exhibit a remarkable difference. Expecting that this would happen again, we decided to choose only one formulation for the in vivo evaluation. Considering the larger size and higher hemolytic effect of sBSANP-F75, and the lower particle yield and lower entrapment efficiency of sBSANP-F40, we selected sBSANP-F62.5 for the in vivo evaluation.

The concentration of SN-38 in vehicle was equivalent to sBSANP-F62.5 (0.315 mg/mL). In Figure 10, however, the measured concentration between F62.5 and SN-38 in vehicle showed a significant difference at the beginning (*t* = 20 min). This could be explained by (1) the crosslinked albumin nanoparticles being immediately recognized by scavenger receptors and removed from the circulatory system; (2) the plasma sample preparation being unable to extract SN-38 within the albumin nanoparticles; and (3) the entrapment efficiency being overestimated by setting $C_{total}$ as the theoretical amount of SN-38 rather than the sum of $C_{free}$ and $C_{extracted}$ although this estimation produced an acceptable standard deviation. 

Table 3 shows that the pharmacokinetic profiles of sBSANP-F62.5 and SN-38 best fit to two compartments, which suggests that the plasma concentration (central compartment) was initially greater than the tissue and, thus, SN-38 moved toward the tissues (peripheral compartment). However, the elimination half-life of sBSANP-F62.5 did not show a significant difference to that of SN-38 in vehicle. This observation can again be explained by the rapid recognition of the crosslinked albumin by scavenger receptors. The extraordinary 19 day half-life was only observed for native albumin molecules [46].

## 5. Conclusions

This study addressed the preparation, characteristics, safety efficacy, and preliminary evaluation of the pharmacokinetic behavior of SN-38-entrapped albumin nanoparticles. From the in vitro experiments, the results demonstrated that the application of sBSANPs has potential as a safe approach for TNBC treatment. In the pharmacokinetic evaluation, although sBSANPs could not elongate the elimination half-life, these nanoparticles were capable of influencing pharmacokinetic parameters. To understand whether sBSANPs accumulate at tumor sites, a breast cancer xenograft model is required, and this study will be underway in the near future. 

## Figures and Tables

**Figure 1 pharmaceutics-11-00569-f001:**
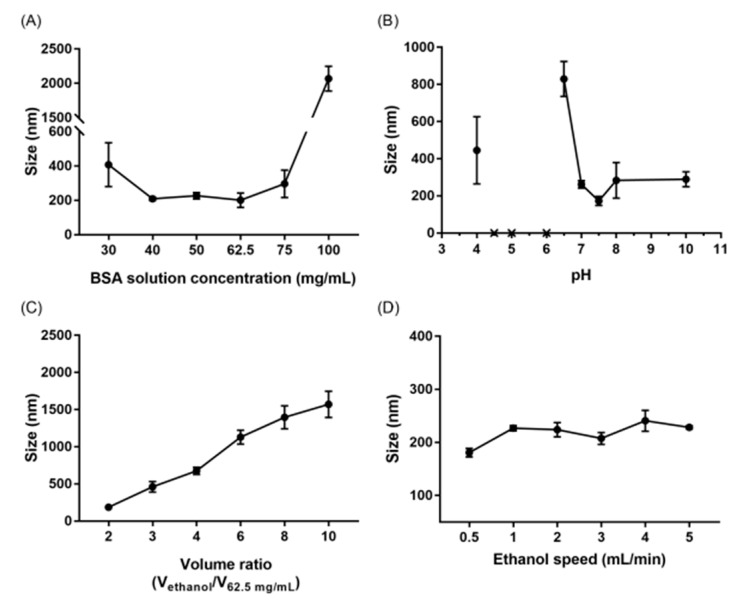
The relationship between bovine serum albumin (BSA) nanoparticles and (**A**) BSA concentrations, (**B**) pH, (**C**) ethanol volume ratio, and (**D**) ethanol addition rate. Note: The crosses in (B) indicate the formulations at pH 6.0, 4.5, and 4.0 precipitated during desolvation.

**Figure 2 pharmaceutics-11-00569-f002:**
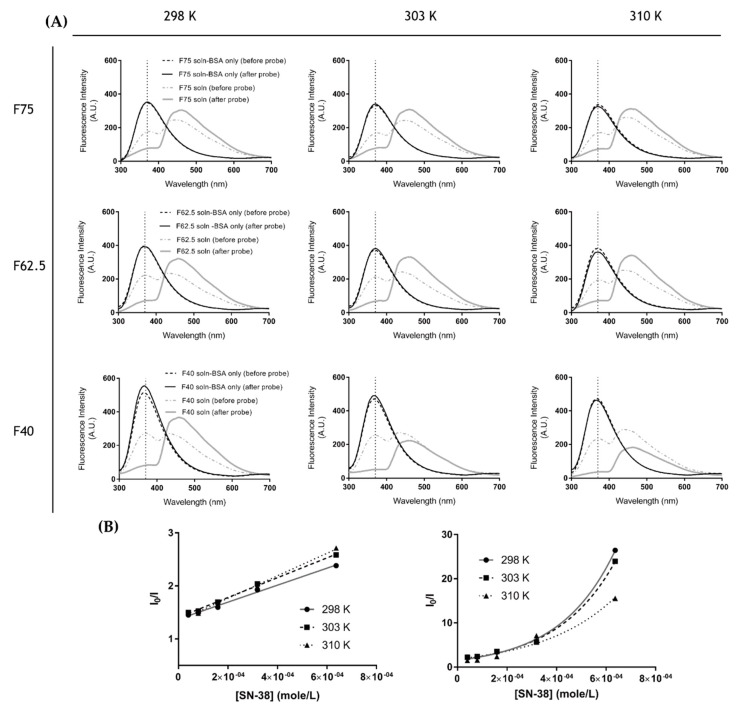
Protein-binding interaction: (**A**) the fluorescence quenching spectra of the BSA solution, and BSA solution with SN-38 (before/after probe sonication); (**B**) the Stern–Volmer plot for the quenching of the BSA solution with different concentrations of SN-38 (before/after probe sonication).

**Figure 3 pharmaceutics-11-00569-f003:**
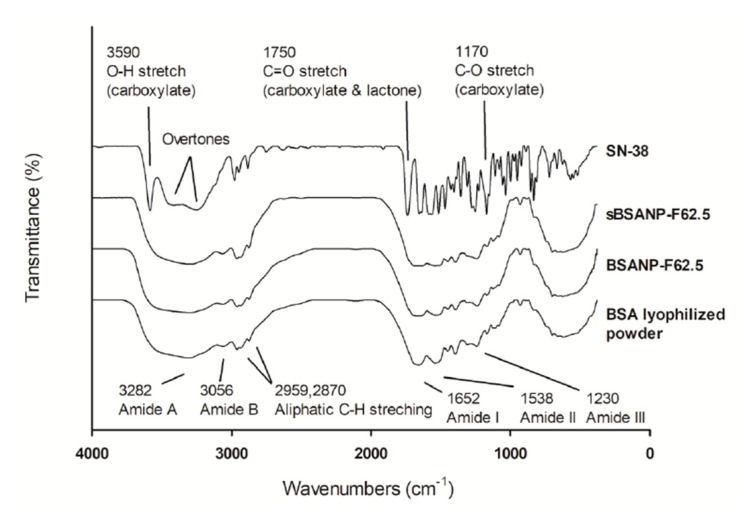
FTIR spectrum of SN-38, sBSANP, BSANP, and BSA lyophilized powder. Abbreviations: FTIR (Fourier-transform infrared spectroscopy); sBSANP (SN-38-loaded bovine serum albumin nanoparticle); BSANP (empty bovine serum albumin nanoparticle).

**Figure 4 pharmaceutics-11-00569-f004:**
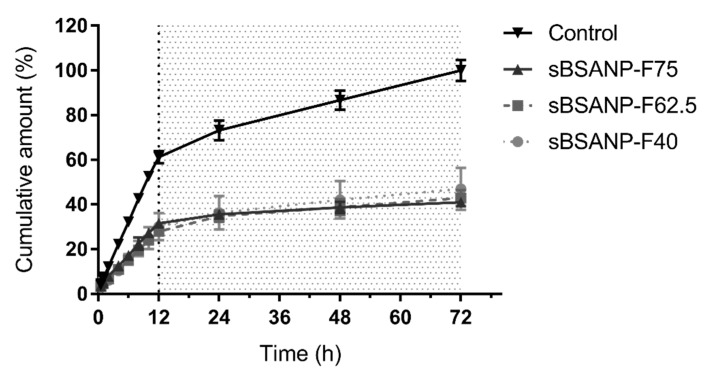
The release profiles of the SN-38 control group and SN-38-loaded sBSANPs with different BSA concentrations at 75, 62.5, and 40 mg/mL of BSA.

**Figure 5 pharmaceutics-11-00569-f005:**
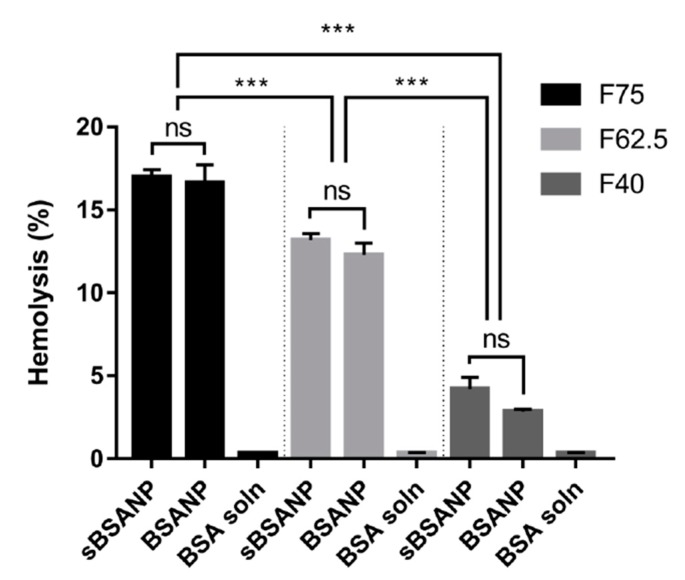
Hemolysis of erythrocytes induced by sBSANP and BSANP formulations at different BSA concentrations. *** *p* < 0.001 was considered statistically significant.

**Figure 6 pharmaceutics-11-00569-f006:**
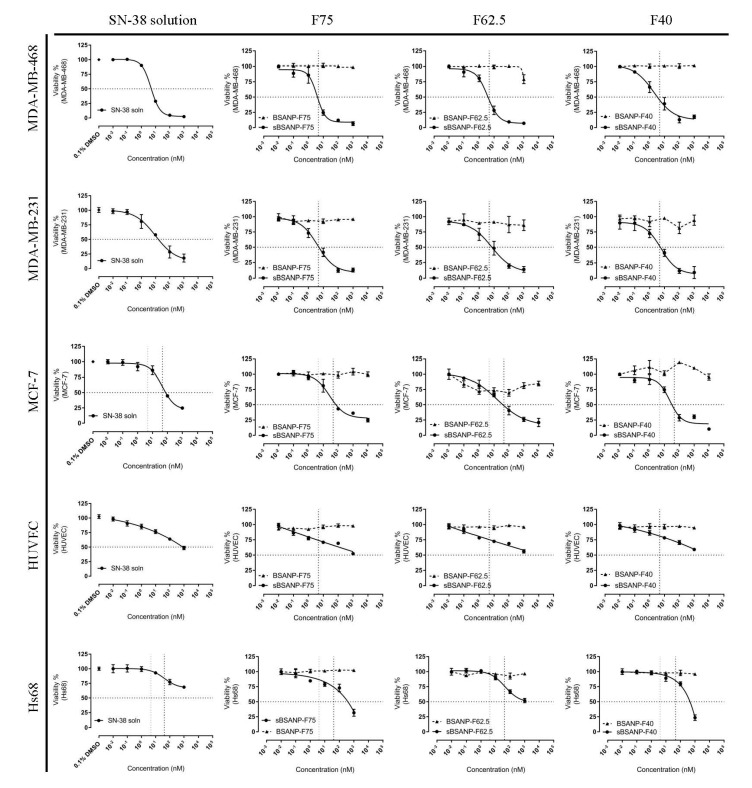
Dose-dependent effect of SN-38-containing control solution, BSANPs, and sBSANPs (F75, F62.5, F40) on different cell lines’ cell viability including MDA-MB-468, MDA-MB-231, MCF-7, HUVEC, and HS 68 cell lines. Note: MDA-MB-468 and MDA-MB-231 (ER−, PR−, HER2), MCF-7(ER+, PR+, HER2−), HUVEC(human umbilical vein endothelial cell), Hs68 (health human fibroblast cell)

**Figure 7 pharmaceutics-11-00569-f007:**
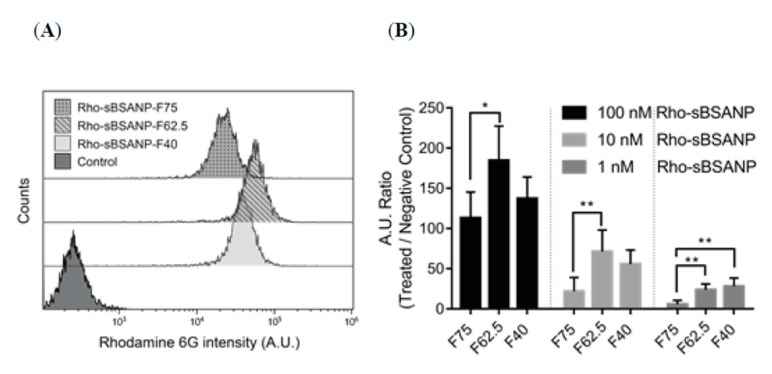
Flow cytometry measurement of MDA-MB-468 treated with Rho-sBSANP formulations for 4 h. (**A**) Number of cells with rhodamine detected and (**B**) qualitative analysis of Rho-sBSANP taken up by MDA-MB-468 at 100, 10, and 1 nM. **p* < 0.05 was considered statistically significant.***p* < 0.01 was considered statistically significant.

**Figure 8 pharmaceutics-11-00569-f008:**
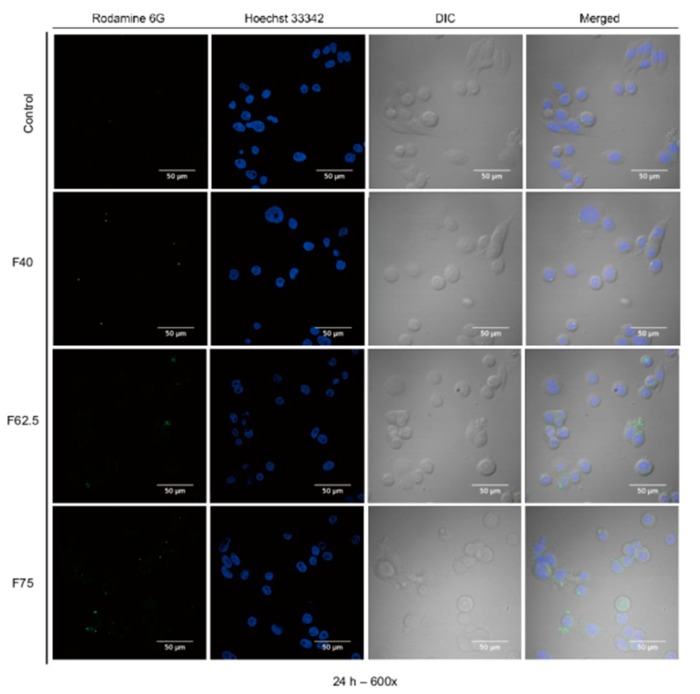
The confocal images of Rho-sBSANP uptake by MDA-MB-468 cells after 24 h treatment. Magnification 600×. Note: Rhodamine 6G channels showing the green fluorescence from Rho-sBSANP distributed; Hoechst 33258 channels showing the blue fluorescence from Hoechst 33258-stained nuclei; DIC channels showing the differential interference contrast; merged channels of channels where liposomes (green) localized at nuclei of MDA-MB-468 cells (blue). Magnification: 600×.

**Figure 9 pharmaceutics-11-00569-f009:**
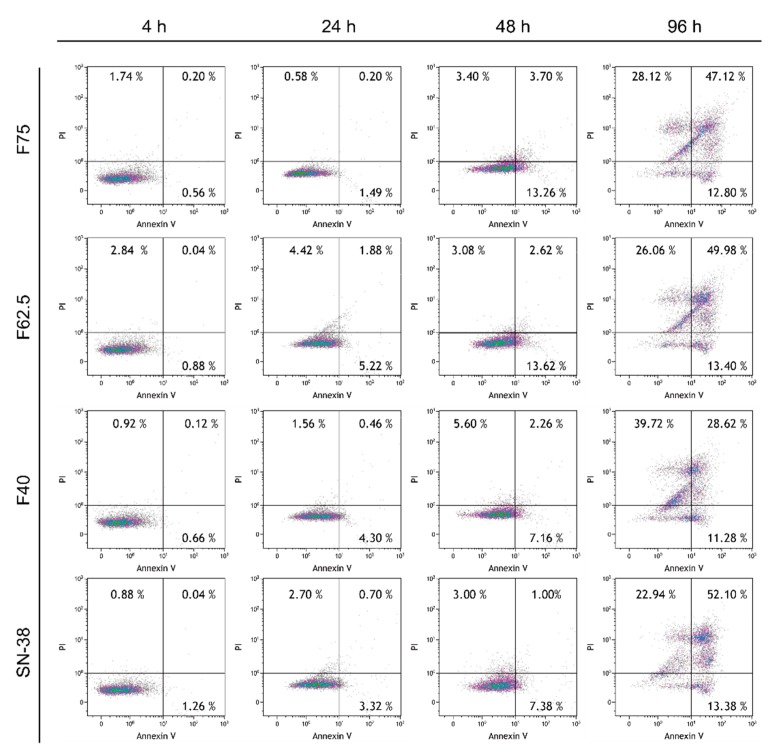
The annexin V–PI apoptosis results after treatment with the SN-38 control and sBSANPs at BSA 40, 62.5, and 75 mg/mL concentrations from 4 to 96 h. Note: The lower left quadrant shows viable cells, the lower right shows early apoptotic cells, the upper right shows late apoptotic cells, and the upper left shows necrosis cells. From 4 to 48 h, the cells moved from the lower left to the lower right quadrant, showing that apoptosis was in progress. At 96 h, cells were observed in the upper right and upper left, indicating that the cells were in late apoptosis or necrosis. The results implied that the SN-38 and sBSANP formulations triggered apoptosis, a desired form of cell death.

**Figure 10 pharmaceutics-11-00569-f010:**
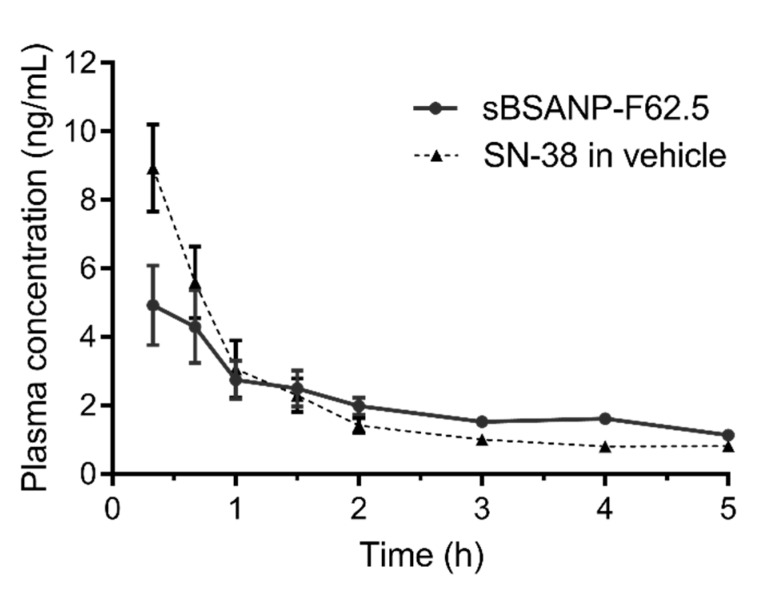
In vivo plasma concentration of SN-38 versus the time profiles of the SN-38 control group and sBSANP F-62.5 at 315 μg/mL in male Sprague Dawley rats after intravenous infusion (0.1 mL/min) administration (n = 4). Note: The SN-38 control group solution was composed of 10% DMSO, 20% Cremophor EL, and 70% PBS (pH 7.4).

**Table 1 pharmaceutics-11-00569-t001:** Characterization of bovine serum albumin nanoparticles (BSANPs) with or without SN-38 in terms of particle size, polydispersity index (PDI), zeta potential, entrapment efficiency, and particle yield.

Formulation	Particle Size (nm)	PDI	Zeta Potential (mV)	Entrapment Efficiency (%)	Particle Yield (%)
**sBSANP-F75**	264.07 ± 13.08	0.23 ± 0.01	−37.16 ± 1.86	71.57 ± 0.84	86.75 ± 1.53
**sBSANP-F62.5**	223.77 ± 2.36	0.21 ± 0.04	−38.30 ± 2.33	63.28 ± 0.62	81.71 ± 5.49
**sBSANP-F40**	134.37 ± 4.48	0.22 ± 0.02	−40.34 ± 0.30	59.14 ± 3.14	65.91± 0.78
**BSANP-F75**	288.23 ± 9.50	0.23 ± 0.06	−41.31 ± 4.76	-	91.63 ± 5.51
**BSANP-F62.5**	202.03 ± 17.30	0.14 ± 0.02	−53.81 ± 3.39	-	85.65 ± 1.43
**BSANP-F40.**	124.47 ± 1.21	0.15 ± 0.04	−53.16 ± 5.33	-	43.02 ± 2.33

Abbreviations: sBSANP (SN-38-loaded bovine serum albumin nanoparticle); BSANP (empty bovine serum albumin nanoparticle).

**Table 2 pharmaceutics-11-00569-t002:** IC_50_ values of SN-38-loaded BSANP at 75, 62.5, and 40 mg/mL BSA concentrations in MDA-MB-468, MDA-MB-231, and MCF-7 cell lines after 96 h treatment.

Cell Lines	IC50 (nM)			
	SN-38 Solution	sBSANP F75	sBSANP F62.5	sBSANP F40
**MDA-MB-468**	4.75 ± 0.17	3.86 ± 0.85	3.65 ± 0.60	2.01 ± 0.67
**MDA-MB-231**	9.93 ± 3.16	4.16 ± 0.75	6.82 ± 1.97	5.72 ± 1.44
**MCF-7**	46.22 ± 9.80	26.92 ± 6.30	15.94 ± 5.96	23.24 ± 6.30

**Table 3 pharmaceutics-11-00569-t003:** Comparison of the pharmacokinetic parameters of the SN-38 control and SN-38-loaded sBSANP-F62.5 mg/mL at 315 μg/mL dosage after intravenous infusion administration to male Sprague Dawley rats.

Pharmacokinetics Parameters	sBSANP-F62.5	SN-38 Control	Significance
Vd (mL)	1643.78 ± 245.46	650.75 ± 64.92	***
CL (mL/h)	444.14 ± 66.61	483.66 ± 80.49	ns
AUC (h × ng/mL)	23.22 ± 3.54	21.19 ± 3.11	ns
t_½_ (h)	7.04 ± 1.38	7.35 ± 2.22	ns
Cmax (ng/mL)	6.29 ± 1.06	15.54 ± 1.70	***
AUMC (h × h × ng/mL)	213.54 ± 67.83	154.92 ± 67.47	ns
MRT (h)	8.91 ± 1.46	7.04 ± 2.57	ns

Notes: *** *p* ≤ 0.001compared with SN-38 solution. Abbreviations: Vd: volume of distribution; Cl: clearance; AUC: area under curve; t_½_: half-life; Cmax: maximum concentration; AUMC: area under the first moment curve; MRT: mean residence time.

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
