# Peer review of "High Potency of SN-38-Loaded Bovine Serum Albumin Nanoparticles Against Triple-Negative Breast Cancer"

_pharmaceutics, 2019, doi:10.3390/pharmaceutics11110569_

Round 1
Reviewer 1 Report
The present paper represents the results of the study of synthesis SN-38-loaded albumin nanoparticles and in vivo evaluation of their potency against triple negative breast cancer.
The experiments were carefully performed and presented in an appropriate way. These results have significant scientific values and level, and I recommend the paper for publication in Pharmaceutics. In spite of this, I suggest the authors rewrite some parts of the manuscript in order to make it easier to follow for the readers. It refers to the sections Results and Discussion.
In my opinion, the section Results can be improved by inserting/replacing some sentences from the Discussion section, which include the direct explanation of experimental design and comments of the obtained results for each particular experiment. On the contrary, all comments and comparison of the obtained results with those from the literature data, supported by the appropriate references, should be left In the section Discussion.
Here are also some detail remarks:
1.section 2.1 - please give the abbreviation for rhodamine when mentioned the first time
2. Section 3.1, Fig. 2 -It is not clear enough for which formulation was performed experiments presented in b, c, d. In general, the comments on the obtained results are missing for all results presented in section 3.
3. It should be interpreted why The release kinetics experiment was performed and what conclusion can be made based on the obtained results, given in Supplementary material.
Author Response
Response to Reviewer 1 Comments
Thank you for reviewing our manuscript and giving many valuable suggestions to us. We had revised the manuscript according to the reviewer’s comments. All the major changes were highlighted in the text of revised manuscript. We hope our responses have adequately addressed the concerns. Thank you for your consideration of this manuscript.
Point 1: In my opinion, the section Results can be improved by inserting/replacing some sentences from the Discussion section, which include the direct explanation of experimental design and comments of the obtained results for each particular experiment. On the contrary, all comments and comparison of the obtained results with those from the literature data, supported by the appropriate references, should be left in the section Discussion.
Response 1: Thank you for reviewer’s valuable suggestion. We had reconstructed the manuscript which had highlighted at the results and discussion section, made reader easier understand.
Point 2: section 2.1 - please give the abbreviation for rhodamine when mentioned the first time
Response 2: We had revised that rhodamine 6G (Rho) at section 2.1 of method section.
Point 3: Section 3.1, Fig. 2 -It is not clear enough for which formulation was performed experiments presented in b, c, d. In general, the comments on the obtained results are missing for all results presented in section 3.
Response 3: Figure 2B shows that the Stern–Volmer plot switched from linear (before probe sonication) to cursive (after probe sonication), that presents two quenching mechanisms were involved after probe sonication, thus strengthening the affinity between SN-38 and albumin molecules. Moreover, we had revised that description of figure 2 at section 3.2.2, made reader easier understand. Figure 2A shows the results of different albumin concentration at 75, 62.5, and 40 mg/mL the albumin solution mixed with SN-38 without probe sonication.
Point 4: It should be interpreted why The release kinetics experiment was performed and what conclusion can be made based on the obtained results, given in Supplementary material.
Response 4: Thank you for reminder. We had addition at section 3.3. In vitro release kinetics of materials and method section. Release profiles and correlation coefficients of SN38 free solution and SN38-loaded sBSANP were investigated, as shown in Supplementary Table 1. The cumulative release percentage data were plotted and fitted based on three mathematical models: zero-order, first-order, Higuchi, Korsmeyer-Peppas and Hixson-Crowell. In addition, the release models of the three formulations fit the zero-order and Hixson–Crowell model best among the five different mathematical models. The release pattern shows that well fit to the zero-order of sBSANP corresponds to a desirable release profile since it suggests that the drug release is independent of concentration and occurs at a constant rate, thus leading to better control of the concentration in clinical applications.

Reviewer 2 Report
Lin et al. present a study characterizing SN-38-loaded bovine serum albumin nanoparticles (sBSANPs). The cytotoxic effects of sBSANPs on a ER+ breast cancer cell line and a triple-negative breast cancer (TNBC) cell line was also evaluated. Although the characterization of sBSANPs is informative in general, the biological significance of the study is lacking. Specific issues and concerns are detailed as follows:
Major concerns:
The authors did not introduce or refer to a previous study (Sepehri et al. , Int. J. Pharm. 2014), where SN38 was loaded with PLGA nanoparticles and tested on breast cancer cells. Accordingly, the results presented in this study is incremental and lacks characterization of in vivo anti-tumor activity, when compared to the previous study. The cytotoxic studies were only carried out on one cell line representing TNBCs. Since the title and conclusions claim effects on TNBC cells, more than one TNBC cell line will need to be tested. It would be better to use normal mammary epithelial cells as a control rather than fibroblasts to show that sBSANPs do not affect the viability of normal mammary epithelial cells. The fluorescence intensities for Rho-sBSANPs should be quantified for confocal images in Figure 8. Is there a correlation between observations in Figures 7 and 8, regarding uptake. In figure 9, vehicle control is lacking and not shown for annexin V-PI apoptosis assays.
Minor concerns:
Line 57, the abbreviation IB should be defined. Line 203, sentence needs to be corrected. In general, it would be beneficial for readers if the results and discussion were combined because as is, the narrative lacks flow and interpretation, especially in the results section. Line 332, the abbreviation Rho needs to be defined during first mention.Author Response
Response to Reviewer 2 Comments
Thank you for reviewing our manuscript and giving many valuable suggestions to us. We had revised the manuscript according to the reviewer’s comments. All the major changes were highlighted in the text of revised manuscript. We hope our responses have adequately addressed the concerns. Thank you for your consideration of this manuscript.
Major concerns:
Point 1: The authors did not introduce or refer to a previous study (Sepehri et al. , Int. J. Pharm. 2014), where SN38 was loaded with PLGA nanoparticles and tested on breast cancer cells.
Response 1: Thank you for your reminding. We had go through the manuscript and reconstructed the present order, especially at discussion section made it more informative. We had addition information about recent approaches, that cited at last paragraph 2 of introduction. In addition, we mentioned that SN-38 against breast cancer by this reference, that discuss at section 4.5. In Vitro Cytotoxicity and Safety Evaluation of discussion.
Reference: Sepehri, N. et al., SN38 polymeric nanoparticles: in vitro cytotoxicity and in vivo antitumor efficacy in xenograft balb/c model with breast cancer versus irinotecan. Int. J. Pharm. 2014, 471, 485-497.
Point 2: Accordingly, the results presented in this study is incremental and lacks characterization of in vivo anti-tumor activity, when compared to the previous study. The cytotoxic studies were only carried out on one cell line representing TNBCs. Since the title and conclusions claim effects on TNBC cells, more than one TNBC cell line will need to be tested.
Response 2: Thank you for encouraging us. Due to higher price of human serum albumin, we selected bovine serum albumin as the pilot materials to understand albumin characterises and optimization factors. Based on the manuscript, we understood the critical process parameter of albumin nanoparticle. Moreover, I totally agree that test more TNBC cell lines would be more convince. However, MDA-MB-468 cells have proven useful tool for investigation of metastasis, migration and proliferation of breast cancer.
We current tried on using human serum albumin as the key materials for anti-tumor activity and the other cell lines of TNBC, it would be worthy.
Point 3: It would be better to use normal mammary epithelial cells as a control rather than fibroblasts to show that sBSANPs do not affect the viability of normal mammary epithelial cells.
Response 3: I agree with test normal mammary epithelial cells to evaluate the safety. In this case of the study, SN-38-loaded bovine serum albumin nanoparticle by intravenous administration route, drug distribution not just focusing around breast. Hence, we selected fibroblasts broader test to understand the safety.
Point 4: The fluorescence intensities for Rho-sBSANPs should be quantified for confocal images in Figure 8.
Response 4: We will quantify the confocal images if necessary as soon as possible. It would take few time to accomplish due to the student graduated.
Point 5: Is there a correlation between observations in Figures 7 and 8, regarding uptake.
Response 5: Figure 7 and 8 present cell uptake by flow cytometry and confocal image method, respectively. Figure 7 presents quantitative results, and Figure 8 could be show the qualitative research method.
Point 6: In figure 9, vehicle control is lacking and not shown for annexin V-PI apoptosis assays.
Response 6: Thank you for reviwer’s valuable comments, we didn’t test the vehicle for annexin V-PI apoptosis assays. But we’d be adding the concept at next SN-38-loaded human serum albumin nanoparticle.
---------------------------------------------------------------------------------------
Minor concerns:
Point 7: Line 57, the abbreviation IB should be defined.
Response 7: Subdomain IB is the marked drug binding region of serum albumin. It has not abbreviation. We had corrected that subdomain made reader more understand.
Ref: RSC Advances 5(61), 2015. DOI: 10.1039/C5RA04395F
Point 8: Line 203, sentence needs to be corrected. In general, it would be beneficial for readers if the results and discussion were combined because as is, the narrative lacks flow and interpretation, especially in the results section.
Response 8: Thank you for the suggestion. We had reconstructed the manuscript made reader easier understand, which had corrected at result and discussion section.
Point 9: Line 332, the abbreviation Rho needs to be defined during first mention.
Response 9: We had corrected at 2.1. Materials.

Round 2
Reviewer 1 Report
I recommend publishing the manuscript in the present form.
Author Response
Dear Reviewer,
Thank you for reviewing our manuscript and give us more valuable suggestion. We had made slightly changes in the manuscript. Please see the attachment. Thank you for your consideration.

Reviewer 2 Report
The authors have made reasonable justification and additions in response to the concerns raised by reviewers.
Some minor typographical errors need to be corrected:
Line 30, the second TNBC cell line should be MDA-MB-231
Line 266, change "his" to "this"
Author Response
Dear Reviewer,
Thank you for reviewing our manuscript and giving many valuable suggestions make the experimental design more construct. We had checked the typing errors in the manuscript, and made some changes in the conclusion section made the meaning more clearly. Thank you for your consideration of this manuscript.
